# The Involvement of Athletes with Intellectual Disability in Community Sports Clubs

Florian Pochstein [1], Gemma Diaz Garolera [2], Sabine Menke [3] and Roy McConkey [4,*]

1 Faculty of Special Needs Education, Ludwigsburg University of Education, 71634 Ludwigsburg, Germany
2 Institut de Recerca Educativa, Departament de Pedagogia, Universitat de Girona, 17004 Girona, Spain
3 Special Olympics Europe Eurasia, Dublin 2, Ireland
4 Institute of Nursing and Health Research, Ulster University, Belfast BT1 6DN, Northern Ireland, UK
* Correspondence: r.mcconkey@ulster.ac.uk

**Abstract:** People with intellectual disability lack opportunities to engage in sports, although the benefits of doing so may be even greater for them. One option is to encourage their inclusion in mainstream sports clubs that exist in nearly all European communities. Although there is a growing knowledge base within organisations such as Special Olympics in adjusting sports to meet the needs of people with intellectual disability, inclusion in community clubs raises additional challenges. This exploratory study aimed to garner the experiences of coaches alongside those of clubs members— with and without disabilities—in 12 community sports clubs in three European countries. In all 20 coaches and 51 members took part in semi-structured interviews. A thematic content analysis was used to devise a conceptual model describing an inclusive sports club. The overarching theme was that inclusive clubs require an ongoing balancing between a focus on sporting skills and performance, with managing the needs and characteristics of the players and the inter-relationships among them. Six subthemes were identified that described the core strategies to the effective functioning of the clubs. However, the vision and commitment of coaches was crucial and their recruitment was the main challenge the clubs faced. In addition, new opportunities for training coaches are needed to support the extension of inclusive clubs across a range of sports and locations. Continuing research could usefully identify the benefits to club members and identify sport-specific adaptations required to make clubs more fully inclusive.

**Keywords:** intellectual disability; inclusion; sports; clubs; coaches; Europe; Special Olympics

## 1. Introduction

World-wide, people with intellectual disabilities (ID) experience social exclusion in different aspects of their life—education, employment, housing and leisure pursuits. Moreover, they often remain the most marginalised internationally [1]. Their social isolation persists despite international declarations expounding their rights to be included in society, such as the United Nations Convention on the Rights of Persons with Disabilities [2] which also includes their inclusion in sport. Article 30.5a states: "*To encourage and promote the participation, to the fullest extent possible, of persons with disabilities in mainstream sporting activities at all levels*".

The participation rate of people with intellectual and other developmental disabilities in sports and physical activity appears to be low internationally [3]. Recent research has identified the implications for their physical and emotional wellbeing. Their social inclusion and for their skills development [4,5]. In past decades, the participation of people with intellectual disabilities in sports has been met within parallel provision such as Paralympics or Special Olympics. The latter is the leading proponent of engaging people with ID in sports [6]. Yet globally, only a small minority of persons with ID currently participate in Special Olympics [7]. The Organisation's Global Strategic Plan [8] re-emphasises the

focus on the social inclusion of people with ID in their local communities: *"an inclusive world for all, driven by the power of sport, through which people with intellectual disabilities live active, healthy and fulfilling lives"*. One outworking of this vision, is to encourage greater participation of children and adults with ID in mainstream sports clubs.

There are several compelling arguments for this approach. Sport Clubs cover a wide range of sports such as swimming, soccer, basketball, judo, golf and tennis and for children through to seniors. In general, clubs are well established and highly recognized platforms for local community activation, engagement and inclusion of people with diverse backgrounds and abilities across Europe in particular. They also offer a neutral entry point to sports for people with ID and their families when some may be reluctant to be associated with Special Olympic clubs as parents are aiming for full inclusion across all areas of life [9].

Special Olympics can struggle to find volunteer coaches and supporters, especially for adults with ID. Hence an option is to encourage those already involved in sports clubs to become involved with athletes with ID through them joining their community club with suitable training and support for coaches that could come from the resources developed by Special Olympics [10]. Through encouraging sports clubs to become more inclusive, further opportunities may also be created for their membership to come from ethnic minorities and recent immigrants, as well as increasing the participation in sports for women [11].

To date there have been two approaches to developing inclusive sports clubs for adult persons with disabilities, although both may co-exist within the same club. Indeed, a similar dichotomy is apparent in other aspects of community life such as employment [12]. First, the community sports club include a section for people with ID as a separate group under the club umbrella, with these members training and competing apart from the non-disabled club members. These teams may affiliate with Special Olympics for competitions.

Second, the community clubs offer inclusive activities where people with and without ID practice sport together on the same teams. Special Olympics developed the concept of Unified Sports by inviting non-disabled players to join with their members in an existing Special Olympics club [13]. A community club reverses this concept by inviting members with ID to join with their non-disabled members on the same team. Competitions can then be arranged by similarly constituted teams within or between community clubs. Even then, in practice, a mix of segregation and inclusion is often common in that during training sessions, athletes with disabilities remain among themselves, while shortly before and during competitions partners without disabilities join in and act together as a team.

Inclusive sports clubs for players with intellectual disabilities is a novel concept, albeit one that mirrors similar initiatives in education for example [14]. The present descriptive study sought to gather the experiences of coaches and mainly adult players with and without intellectual disabilities who were involved in inclusive sports clubs.

*Aims and Objectives of the Study*

The overall aim was to gain an understanding of how successful inclusive clubs operate so as to assist with their expansion. To that end, three objectives were identified.

1. To scope the attitudes and experiences of coaches and non-disabled partners to the inclusion of people with ID in sports generally and in their local clubs.
2. To obtain the experiences of athletes who took part in these clubs.
3. To understand the facilitators and barriers to promoting inclusive practices for athletes with ID in mainstream sports clubs.

In order to ensure greater validity, the study was undertaken in three European countries and involved a variety of sports clubs that adopted the two approaches described above.

Ethical approval for the study was obtained from Ulster University, UK. (Ref: FCNUR-20-004).

## 2. Method

Special Olympics (SO) personnel identified three countries in which athletes with ID were included in mainstream clubs. Local SO personnel in Germany, Spain, and Belgium identified the community clubs personally known to them who had enrolled members with ID. In each country, an independent university partner was recruited to assist with the information gathering from the club personnel. In particular, they had experience of interviewing people with ID [15]. The steering group for the project consisted of the Special Olympics national representatives and their research partner plus the two PIs (RMC and SM). This group met by Zoom to agree the information gathering approaches, the transcription of the interviews and data analysis.

### 2.1. Participants

Special Olympics personnel personally contacted each club to recruit their participation in the study. Of the 12 clubs identified, seven were Unified Sports Clubs and five had a segregated stream within their club. A wide range of sports were offered including soccer, handball, basketball, judo, swimming, athletics and canoeing. The number of members in the clubs ranged from 6 (canoeing) to 240 (various sports) with a median of 30. Their ages ranged 10 to 70 years of age.

An information leaflet about the project was distributed by clubs to their coaches and members with details of the consent procedures: in particular assurances of confidentiality were given and the option to withdraw at any time. The contact details of the 12 clubs who had agreed to take part were passed on to the university personnel who arranged suitable times for the interviews with the nominated personnel who had self-selected to participate. No exclusion criteria were applied.

In all, 20 coaches (11 male and 9 female) were interviewed: 11 Germany, 5 Spain and 4 Belgium. They came from a range of backgrounds including sports sciences, former elite athletes, therapy, special education and parenting. The length of time they had been coaching ranged from one to 30+ years with a median age of 10 years.

Of the 32 athletes who participated, 22 were males and 10 females with ages ranging from 18 to 49 years and a median of 21 years.

A total of 19 non-disabled members were interviewed, 6 males and 13 females with ages ranging from 21 to 67 and a median age of 21 years.

No further demographic details were requested from respondents so as to maintain their anonymity.

### 2.2. Procedures

As this was a descriptive, exploratory study, semi-structured interviews were chosen to gain insights into the participants' experiences of the inclusive clubs. The Steering group developed probe questions in line with the aims of the study and informed by past studies of Special Olympics Unified Sports and Inclusive clubs [16,17]. This ensured that a uniform approach was adopted across the three countries with questions translated into German, Spanish and Flemish by the university partners.

In Germany and Belgium, the interviews were conducted by university personnel face-to-face as individual interviews with the coaches and group interviews with the disabled and non-disabled members. In Spain the interviews were conducted by Zoom due to continuing COVID restrictions. The interviews mainly took place during or after training sessions at the Clubs.

The interviews were conducted in May to June 2022. The interviewers first recapped the purpose of the study and outlined the consent procedures. Verbal consents were obtained along with permission to make an audio-recording of the interview. Informants were then asked to provide details of their club.

With coaches the questions covered how they became involved in coaching athletes with intellectual disabilities in their club: why did they think it was important to have athletes with intellectual disabilities playing sports; what were the three main success

factors for including members with intellectual disabilities; what were the three main challenges; how have the members reacted to one another; in what ways have you as a coach or other coaches, benefited; how would you like to see your club develop in the future? Finally demographic details as noted above were gathered. The interviews lasted on average 20 min.

A similar process was held with club members. The probing questions included: how did you become involved in doing sports in your club: what is the best part of being in this club; have there been any negative experiences or reactions; how do the members get on with each other; what have you learnt from being in your club; why did you think it is important to have members with intellectual disabilities playing sports? Do you have any other particular memories or special experiences that you would like to tell us about? The individual interviews lasted on average 12 min with group interviews taking up to 20 min.

*2.3. Data Analysis*

The interviewers made a verbatim transcript of the interviews in the local language. These were translated into English by the bi-lingual university researchers. A thematic content analysis was undertaken using the six steps proposed by Braun and Clarke [18]. The subthemes were initially identified by two authors working separately on the transcripts from Germany (FP) and from Spain/Belgium (RMC). The analysis was carried out across all the transcripts of coaches and club members as a form of data triangulation. It was evident that data saturation had been achieved as no new themes emerged in the analysis of later transcripts.

Through discussion between the two authors, a combined list of subthemes was created from which the super-ordinate themes were identified and cross-checked with a third author (GDG) who had conducted the interviews in Spain.

The findings are presented in two sections. First the strategies that made for successful inclusive clubs and second, the main challenges facing coaches of inclusive clubs.

**3. Results**

The overarching theme to emerge from the interviews was that inclusive clubs require a balance between a focus on sporting skills and performance with managing the needs and characteristics of the players and the inter-relationships among them. Figure 1 summarises the subthemes within the two domains of sports and of players.

**Figure 1.** The domains and subthemes in creating inclusive clubs.

*3.1. Creating Inclusive Clubs*

Arguably the two domains—Sports and Players—are present in all sporting clubs and activities but regular sports clubs may place greater attention on the sports whereas in segregated disability sports the focus arguably is more on the players and meeting their needs, with sports being a medium for doing this, primarily as a recreational activity [19]. However our data suggests that inclusive clubs require an ongoing rebalancing of these domains and using quotations from coaches and players insights were gained as to how

this is achieved. (Note: the codes are B-Belgium; G-Germany; S-Spain. A-member with ID; C-Coach; P-non-disabled member).

Two Spanish football coaches described the balanced they sought with their inclusive clubs.

> *Apart from the training in sport, it is very important to know the people. So that is one of the strongest points that perhaps as a club we have. The coaches know the people and when it comes to adapting the exercises, sports training is very important, but it is also very important to know the people. (S02C)*

> *(We) play as a team, that is, to get people to integrate and play as a team, to know how to kick a ball, to know how to pass the ball to each other and not get angry, you know? . . . in the football school, we are another family, you know who they are, you give them affection, you give them motivation, you tell them how good they are, how well they do it. (S03C)*

A player with intellectual disability also spoke of the duality which was echoed by two non-disabled members.

> *The really good thing is being able to compete. To be able to play in different championships. And to be able to be with your teammates, to be able to make new friends, a bit of sociability. You can enjoy yourself. Above all, having fun. (S01A)*

> *It has allowed me to learn first-hand the importance of connecting with people and improving our skills to do so. Just like for people without disabilities, sport provides countless physical and psychological benefits to those who practice it. (S04P)*

> *People have already said to me, why I'm not in a sports group, where performance or something like that plays a role, ... I could also do that—train boys in football, where it's all about performance. But a lot of people just don't understand that it's not just about that in sport, but that fun and togetherness is also important. It has allowed me to learn first-hand the importance of connecting with people and improving our skills to do so. (G06P)*

*3.2. The Sports Domain*

Interviewees elaborated on how sport skills and performance were developed. The three subthemes under sports (training, support and programme design) come from the framework proposed by Geidne and Jerlinder [20] in their literature review of research into the inclusion of children with intellectual disabilities in sport. Our analysis confirmed its suitability with adult persons in community settings.

3.2.1. Training

The coaches described the approaches they used in training club members. The need to personalise the training regime was emphasised, allied with careful assessment of the members' current level of proficiency.

> *Well, first we develop levels. The players don't all have the same level. What happens when someone doesn't have the same level? He gets very frustrated, he suffers a lot of frustration for never being able to reach the level of someone else with a much higher level. We try to make sure that everyone plays at his level without despising the moment when they all play together. (S03C)*

> *At the same time, when my son was about ten, eleven or twelve years old, I noticed that the gap between him and the rest of the group was getting bigger and bigger. At that time he was still in the regular athletics group. Then came his own realisation, "I can't do all of this as well as the others." And then I thought to myself, "Before he gets frustrated, and quits everything, and doesn't want to do anything anymore, we have to act somehow." And so the idea just came more and more: We have to create a sports programme, where they simply feel comfortable, where you can cater to their needs. (G01C)*

*Knowing them well is what makes you decide the exercise you have to do with each person in a training session, because at the end of the day, training does not mean that we are going to do the same exercise with everyone, perhaps yes, but with different adaptations, with each one of them, especially people with more or higher needs, above all in the area of attention and concentration, perhaps. (S02C)*

The members also recognised the importance of training for ensuring their participation in inclusive teams.

*If you train a lot, it will pay off; if you don't train at all, you won't give what you have to give in the match. That's why I mean that if you, for example, train every day and give your best in every training session, then you'll do that in the match, and you can have a good match. Or you can do badly in a match, depending on what you have given of yourself in the training sessions. (S01A)*

But the training also extended beyond the sports field as these coaches identified.

*They experience an incredible ability to work in a team, which also develops, when they play against each other in relay games, and cheer each other on. They do create personal links between them, either through WhatsApp or social networks, they maintain the relationship outside. And those who have the capacity and the possibility, they carry out activities among themselves on their own. (G03C)*

*I also want the guys to achieve independence. I want the people in my team to be able to move around on public transport. I help them at the beginning, but if we are going to play on a football pitch—they have to know how to get to that football pitch, they have to know how to get to the place where we train. The unity of the team, transport, autonomy, commitment, that they learn to have a commitment to our team. (S03C)*

### 3.2.2. Support

The second subtheme related to the supports that coaches and members offered to one another.

*Maybe the fact that we can pay a bit more attention to performance, that is, we can assess it and say "let's run another lap" or "go a little faster" or "look at your posture", but on the other hand also the normal interaction, the points of contact between, I'll call it "the normal world" and "their world". That's actually exactly how it works, so I don't train with my athletes any differently than I would do with other young people. (G03P)*

*I am in other sport clubs, and I believe that there is more fellowship and they help each other a lot more in this club (Special Olympics football). (S01P)*

*I like the good vibes that exist between all the teammates, which makes you feel integrated with everyone. (S02P)*

*You win people over by talking nicely rather than shouting. OK, when a player or a person does something that we think they shouldn't do, you have to explain why, but you don't have to shout at them or even scream at them. (S03C)*

As Geidne and Jerlinder [20] noted, support can either be provided directly to individuals as shown in the quotes above, or by enlisting the support of others, such as family members and non-disabled members, as these German coaches describe.

*Without the support of the parents, the training would not be feasible for us. The athletes are often not able to come to training alone, mobility is usually very limited. Many cannot drive a car, and public transportation is poor. Without the help of parents and friends who drive our boys to training, it wouldn't work. (G01C)*

*In the meantime, carpools have formed. The athletes without disabilities pick up the athletes with disabilities by car and bring them back after the training. (G03C)*

### 3.2.3. Design

The third subtheme related to overall design of the sports training and the adaptations that may be necessary to accommodate members with ID.

*(In judo), the treatment is also the same for ID athletes than that of the regulars. The people who go against each other are pitted against each other in terms of age and the same bond. The chance is that ID-judokas and regular judokas can play sports together and have a good relationship with the people with intellectual disabilities. (B04C)*

*I learned in my coaching courses that every player needs a ball and that you do technique and practice that and that it's built up and so on. I do it sometimes like this, we warm up and then we kick (play soccer) for a while. Then we kick. That's just what they want. And there's everything there, there's technique, conditioning, everything there. Sometimes we do special technique or conditioning, but quite often we just do it so that we play. (G02C)*

*The participation of disabled and non-disabled athletes is the best example of the possibilities offered by sport as a tool for social integration. (S02P)*

*Even if I don't succeed, at least let me try—"I haven't achieved it this time, but maybe next time I will!". And we always have to keep on going up, you know? That's what our objective is. (S03C)*

### 3.3. The Players Domain

Alongside describing their sporting experiences, the interviewees elaborated on their personal development and the inter-relationships among members and among coaches and members with and without ID.

### 3.3.1. Relationships

The members spoke of the friendships that had developed and which extended outside of the club.

*(I like) the coaches and the companionship with the other athletes. Making friends and meeting new people. I've learnt to improve my relationships with my teammates. (S02A)*

*But what has actually happened now, what we didn't have in mind at the beginning, is that amazing friendships have developed. (G01C)*

*I loved to compete because of the possibility to meet people with the same interests and enjoy sport together. (The relationship) is wonderful on both sides. It is very close, respectful and full of affection. (S04P)*

*I just really enjoy being here, in this sport. But I also really enjoy letting off steam. Yeah, and I have a lot of friends here. Yes, and not only that, not only many friends, but I also have a boyfriend. (G02A)*

*They (non-disabled members) are told that if they see that with one person they connect more and they feel like sharing time with him or her outside the sports school, then they have the right to do so. (Name) who is a volunteer, and a disabled athlete got on very well and started to establish a good bond, and they meet up on their own and go for a drink. They go to the swimming pool, they spend time together. (S01C)*

The development of relationships opened further opportunities for members with ID as this Spanish coach reported.

*Both individual and team sports can be good for social relationships. This is what we are here for! We have Paralympic athletes in athletics in a high performance group in Madrid. And they are as included there as if they were in a team sport. (S05C)*

### 3.3.2. Emotions

The interviewees described the emotions that were engendered in inclusive clubs. These fuelled their motivation to train and take part in competitions. Coaches also needed to be sensitive to negative emotions of frustration and anger.

*It gives me a lot of things—the values and the fellowship..., but above all, it gives me the chance to grow as a person. The hugs, companionship and joy that is shared among the athletes every day. (S04P)*

*I often had children, who came to me with pent-up aggression, and they took it out on others. But now, when they have been with me for a while, I notice, how they really have it under control, that the sport takes that energy out of them, and that the aggression is no longer directed against others. (G05C)*

*I like to compete because of the great emotions generated and the companionship we generate during competitions with the teammates. . . . When we have a good match, when we do it great, the atmosphere is incredible . . . we all love it and enjoy it a lot. (S01P)*

*Yeah, you get a lot back as a partner, so when you—I always come straight from school— when I come, you're usually stressed out or not in a good mood, and afterwards I'm always in a good mood because you just get that back. (G05P)*

*We are all volunteers here, we do this for pure pleasure. We like what we do, we feel good with what we do, it is our leisure, this is our leisure. I see the happiness with which they come to play and they are delighted, they are looking forward to it. (S03C)*

*The objectives that we have? To have fun. We are here to do sport, to give the kids a chance to do certain camps like this is one (that) we are going to have for three days, we will share breakfasts, lunches, dinners, laughs . . . Well, that's what Unified football is like. Basically, we don't just play football. (S03C)*

*Sport is very beautiful, but one has to win, and all the others have to lose. That's how it is. So failure is something that some of them don't give importance to, but there are others who do have that feeling of failure. We work a lot on the fact that we do this (sports) for fun, that it's something positive for them, that not everyone can win. (S01C)*

### 3.3.3. Values

The third subtheme spoke to the values that underpinned the participation of coaches and non-disabled members in inclusive clubs.

*What I have learnt the most, is to change my vision of sport. To see beyond the physical aspects. Because normally, when we do sport, we focus on the physical, on health, more on a physical level. And working here has given me the opportunity to see sport from other areas, such as the social aspect. And also seeing sport, as I said before, as a resource to gain autonomy. (S05C)*

*And for example, the one trainer who is also here now, that is a friend of mine and she had, she has not been here that long, for that she had no contact at all with people with intellectual disabilities. And then she was a bit more distanced and now she's just with us and thinks it's mega great and has a completely different perspective on it (G04C)*

*Some have their limitations, some can run more, others can run less, some are taller or shorter, some can control the ball well, others have to go a little bit more slowly, others can perhaps play defence, others play well as goalkeepers... In other words, we all have our limitations. (S01A)*

*Well, the athletics club had club championships here and then we did an inclusion relay with the boys. Four times a hundred metres, with two from our club and two from the BSG (names club). In the end, everyone won, so no one left the field sad. Everyone cheered like hell. If I have a competition myself and don't reach my time, then I'm sad, but I learned from them to just be happy. You made it and you finished, so you can be happy. (G01P)*

*In the club we no longer talk about special basketball but just the basketball because we believe that everyone is equal. And especially that the coaches have carried out their utmost to train them. (B03C)*

*I don't care if they win or lose, but I want them to make an effort. And I know that sport is good for other things, not just to win a medal. Sport brings the value of effort, perseverance, and is a means to develop other life skills. (S05C)*

*3.4. The Challenges Facing Inclusive Clubs*

It was the coaches who spoke mainly of the challenges they faced in sustaining an inclusive membership. Chief of which was having more coaches and suitable non-disabled team members. However more funding was required to cover membership fees of players from low income families and to enable participation in competitions.

*I think that our club already offers and has a lot, in the running group we have a lot of coaches, but in the basketball group, for example, we still lack people to join. I think there are only two coaches and the rest are athletes. I would like to see this expand and not just have a basketball group without a disability and one with a disability, but for people to want inclusion and try it out. Because they can play just as well and why not together? So that there are more and more mixed groups. I often don't understand why people separate them. (G08C)*

*If there were three coaches at the training sessions, we could see many more things. We could evaluate many more aspects of training and we could improve them. One could deal more with the physical issue, one more with the technical aspects, and one more with the tactical aspect. (S05C)*

*If we could have a physio and also a nutritionist, that would be very welcome. Because we don't have that. (S04C)*

*I would like to see us have a small league with four teams or so, and it would be really nice if there were a lot more teams here. (G07C)*

*To be sustainable so that we can maintain the human team. In the end, if you have a good coaches team, you can do many things. So that would be the first thing: stability in human resources. And also to be able to buy more adequate material. And with all this, in the end, we would have a higher quality in the activities that are carried out. That would be the essential things. (S04C)*

*The partners we ask to participate in a Unified sport should have enough experience to guide/coach those people and you do not have that with younger people. (B03C)*

A Spanish coach felt sports federations could be more sympathetic towards players with ID and support for inclusive clubs.

*I have had some negative experience with the federations that control this whole world. I think that many times we put money above many things, you know? And sometimes these associations don't understand the kids. They don't understand their day to day life, their world or what it's like. So yes, I do miss them being a bit more empathetic. (S03C)*

The need for coach training was also highlighted, especially to enable gifted sports players with ID to move on and also to make other sports clubs more inclusive.

*When someone stands out, we look for ordinary clubs where they can train. And we refer them there. And this is good. We should do this with more sports. (We could) offer this training to the coaches who are going to be with them . . . It is not a formal training, but one of us goes the first day, the second day, the third day, and as many days as it takes to see if the athlete is really integrated into the training group. And then we also evaluate with the family how the athlete is doing with the other coach, with the other athletes in the group. And if he/she is fine, then he/she continues there. And if not, we try to find another inclusive resource for him/her. (S05C)*

## 4. Discussion

This study has a number of strengths. Rich qualitative details were obtained of the lived experience of coaches and members of inclusive clubs across a range of sports drawn

from three European countries. A conceptual model was proposed of the key elements that contributed to successful inclusion of adult persons with ID into community sports clubs. This extends the insights gained from studies that have focussed on the participation of children and youth with ID in mainstream sports settings [20] and from Unified Sports promoted by Special Olympics [21].

However, knowing what is needed to make community sports clubs more inclusive, does not guarantee it will easily happen. On the contrary, our experience suggests that the enthusiasm and commitment of the coaches we interviewed were vital in developing and sustaining an inclusive ethos in their clubs. Equally they noted the need to recruit more coaches: in part to create a better experience for the club members but to provide continuity when existing coaches 'retire'. However, the 'job description' for inclusive coaches has to go beyond the training that is commonly provided for coaches in specific sports [10]. In essence it must also focus on the people domain described in the model and which seems to be applicable across different sports and what has been referred to as "the social-relational model" in coach education [22]. Sharing this form of people knowledge and insights could come from experienced coaches, through training others at a personal level as was described above. However, such an approach is time consuming and makes further demands on what is likely to be a small cadre of coach trainers.

A more feasible approach would involve the development of multi-media training modules that guided coaches in fostering the personal development of players and their relationships with other players as described in this study [23]. These modules might be developed by Special Olympics for use within their own organisation but made available to other providers of training for coaches. A parallel approach would be for Sports Federations to commission inclusive training materials for their coaches. Better still, might be a partnership approach between Special Olympics and international sports federations.

However, the training needs to be rooted in practice rather than theory and aimed at trainees with some experience of people with ID. In this respect an obvious target group could be the members of inclusive clubs—both disabled and non-disabled—to develop their role and skills as coaches. For example, Special Olympics Germany offers training as an exercise instructor assistant for people with ID for this purpose. In addition, other potential coaches are family members of people with ID and professionals working in support services for people with ID with an active interest and experience of playing sports. These have been an invaluable source of volunteer coaches for Special Olympics albeit that the role of an inclusive coach is different in style and location.

Inclusive sports clubs would also benefit from having the support of others, notably non-disabled members and the recruitment of members with ID with an interest and a talent in particular sports. With the latter, the support of parents and siblings may need to be nurtured. These groups would benefit from awareness raising activities—along the lines of YouTube and social media videos—but supplemented by personal contact with members of inclusive clubs [24].

The dearth of research into sports and people with ID has often been lamented [25] and further investigations could overcome some of the limitations of this study. For example, further research could usefully explore the variation in inclusive practices required by different sports which this study was unable to do due to limited numbers of informants in the specific sports. Similarly, the outcomes from the two approaches to inclusive sports could be compared. Those that had a separate strand for players with ID and those that operated on an integrated approach [12]. Longitudinal studies could be undertaken into the establishment of inclusive clubs and the factors that contributed to sporting performance and personal development of members as well as the sustainability of the clubs. More attention may need to be paid to the characteristics of players with ID who thrive in inclusive environments and those who do not.

Nevertheless, the practical lessons even from this small-scale study are clear. Athletes with ID can be accommodated within mainstream clubs provided coaches are equipped with the sports expertise and social skills to manage the extra demands placed on them.

Moreover, the non-disabled club members also stand to benefit from the experience which should provide further encouragement to clubs to make their membership more inclusive.

## 5. Conclusions

The rhetoric around inclusion in sports—both recreation and competitive—is well established in policy statements. However, translating these aspirations in reality has barely begun for people with ID. This study provides hope that inclusive sports clubs can be created to the benefit of their participants. Moreover, the strategies underpinning their success have started to be identified. In contrast with policies that emanate from governments and sporting federations, action has to start from the grassroots, namely the members of sports clubs that exist in every community. That is the most hopeful aspect of translating words into actions.

**Author Contributions:** Conceptualization, S.M. and RM.; methodology, F.P., G.D.G., R.M. and S.M.; formal analysis, R.M. and F.P.; investigation, F.P. and G.D.G.; resources, R.M. and S.M.; data curation, R.M.; writing—original draft preparation, R.M.; writing—review and editing, R.M., F.P., G.D.G. and S.M.; project administration, R.M.; funding acquisition, S.M. All authors have read and agreed to the published version of the manuscript.

**Funding:** This research received no external funding.

**Institutional Review Board Statement:** The study was conducted according to the guidelines of the Declaration of Helsinki, and approved by the School of Nursing Filter Committee, Ulster University, Northern Ireland. Ref: FCNUR-20-004.

**Informed Consent Statement:** Informed consent was obtained from all participants involved in the study and assurances were given about the confidentiality of the information they provided.

**Data Availability Statement:** The data reported in this paper is available on reasonable requests to the corresponding author.

**Acknowledgments:** Our thanks to Special Olympics personnel: Jenny Wolf in Germany; Álvaro Terreros Martínez in Spain, and Hendrik Desmet in Belgium who facilitated contact with the clubs and to the coaches and members who took part in the interviews.

**Conflicts of Interest:** S.M. is an employee of Special Olympics Europe/Eurasia. The remaining authors declare that the research was conducted in the absence of any commercial or financial relationships that could be construed as a potential conflict of interest.

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
