# Peer review of "The Involvement of Athletes with Intellectual Disability in Community Sports Clubs"

_disabilities, doi:10.3390/disabilities3010005_

Round 1
Reviewer 1 Report
I am glad I had a chance of reviewing this paper, and I hope my comments will help to enhance and broaden the perspective. The study is innovative and addresses important information on the area of athletes with intellectual disability well-being. Overall, this is a well written article. Even though, the manuscripts present some flaws that must be considered. I recommend its publication after minor changes. Some additional details could be added to the method section, below you will find my specific recommendations.
1)Materials and methods/This section needs to be described a little more, such as what the inclusion and exclusion criteria were.
2)Discussions/Point out the strengths and limitations of your study.
3)The general objective and specific objectives should appear at the end of the introduction.
4)The sample is very small for this type of study that should pay careful attention to its inference results, and should be limited in the article.
5) It should be indicated whether the ethics committee that approved the study (or informed consent form. ICF)
6) I recommend adding one or two more specific conclusion to highlight your main results of study.
Reviewer 2 Report
The study entitled:” The involvement of athletes with intellectual disability in community sports clubs” aimed to gain an understanding of how successful inclusive clubs operate so as to assist with their expansion.
This study has several objectives. The first aim is to scope the attitudes and experiences of coaches and non-disabled partners to the inclusion of people with ID in sports generally and in their local clubs. The second aim is to obtain the experiences of athletes who took part in these clubs. The third aim is to understand the facilitators and barriers to promoting inclusive practices for athletes with IDD in mainstream sports clubs.
The research has scientific relevance. The article was written very well. The findings were novel. However, authors need provide some explanations and answer the questions below in order to consider your study for future publication.
- Intellectual disability is specifically focused on one disorder which is concerned with intellectual and adaptive functioning. On the other hand, the term developmental disability encompasses people with intellectual disabilities but also includes physical disabilities. In your study, sometimes you mention ID and others times you use IDD. Please, harmonize that throughout the manuscript.
- Describe further the particularities of participants with an intellectual disability.
- I would like to have an explanation concerning the choice of countries: Why these 3 European countries? is there a particular specificity compared to other European countries?
- You mention that : “The length of time they had been coaching ranged from one to 30+ years with a median age of 10 years.” : Have you noticed any differences in the coaches' discourse, which changes depending on their expertise?
- “Of the 32 athletes who participated, 22 were males and 10 females”. Is gender difference has been considered in your study? Justify
- In all, 20 coaches were interviewed (11 Germany, 5 Spain and 4 Belgium) : Have you noticed any country-related differences?
- Please give the coaches’ gender. Is coaches’ gender has been considered in your study?
Reviewer 3 Report
The study entitled "The participation of athletes with intellectual disabilities in community sports clubs" aims to understand how successful inclusive clubs work to help their expansion.
The research is novel and highly practical for developing policies and strategies for athletes with intellectual disabilities.
However, the authors should incorporate some minor aspects:
line 84: Incorporates a hypothesis
line 99: Incorporates the code of ethics committee
Results: Add a table to summarize the results
Discussion
Incorporate a subheading with practical applications
Could you mention the limitations of the study?
